- Physical and biological processes driving seasonal variability of Nitrate
- budget and biological productivity in the Congolese upwelling system
- 3 Landry Junior Mbang Essome<sup>4,3</sup>, Gaël Alory<sup>3</sup>, Casimir Yelognissé Da-
- 4 Allada<sup>1,4,5</sup>, Isabelle Dadou<sup>3</sup>, Roy Dorgeless Ngakala<sup>1,2</sup>, Guillaume Morvan<sup>3</sup>
- <sup>1</sup>Department of Oceanography and Applications, International Chair in Mathematical Physics
- 6 and Applications, University of Abomey-Calavi, Cotonou, Benin.
- <sup>2</sup>Department of Oceanography and Environment, Institut National de Recherche en Sciences
- Exactes et Naturelles, Pointe-Noire, Congo.
- <sup>3</sup>Université de Toulouse, LEGOS (CNES/CNRS/IRD/UT), Toulouse, France.
- <sup>4</sup>Laboratoire de Géosciences, de l'Environnement et Applications, Université Nationale des
- Sciences Technologies, Ingénierie et Mathématiques, Abomey, Benin.
- <sup>5</sup>Laboratoire d'Hydrologie Marine et Côtière, Institut de Recherches Halieutiques et
- Océanologiques du Bénin, Cotonou, Bénin.
- Corresponding author: L.J. Mbang Essome (<u>landrymbangessome@gmail.com</u>)

21

27

31

# https://doi.org/10.5194/egusphere-2025-5112 Preprint. Discussion started: 11 November 2025 © Author(s) 2025. CC BY 4.0 License.

Acronyms:

CTW: Coastally trapped waves

CUS: Congolese Upwelling System

EKW: Equatorial Kelvin Waves

GG: Gulf of Guinea

MLD: Mixed Layer Depth

SST: Sea Surface Temperature

SLA: Sea Level Anomaly

CHLa: Chlorophyll-a

EBUS: Eastern Boundary Upwelling System

TAUS: Tropical Angolan Upwelling System

Abstract

68

69

71

82

87

89

92 The Congolese upwelling system, located in the southeastern Gulf of Guinea, is a highly productive marine ecosystem influenced by both local and remote physical forcing. This study investigates the seasonal variability of the nitrate budget and biological productivity in this region using a highresolution (1/36°) coupled physical-biogeochemical simulation with the NEMO-PISCES model. The analysis highlights the relative contributions of physical and biological processes in modulating nitrate concentrations in both the mixed layer and the euphotic zone. Results reveal a semi-annual cycle of nitrate, with two upwelling periods (May-August and December) and two downwelling periods (January-April and October-November). These cycles are primarily driven by the passage of coastal trapped waves forced by equatorial Kelvin waves, inducing vertical thermocline displacements and regulating nitrate availability in the euphotic zone. The nitrate budget analysis shows that the vertical advection, linked to the coastal trapped waves (CTWs), is the dominant process supplying nitrate to the mixed layer during the main upwelling season. However, near the Congo River mouth (5.5°S-6°S), the horizontal advection plays a key role, supplying significant amounts of nitrate through the river plume. In the lower euphotic layer, the vertical mixing contributes to the nitrate loss during the upwelling but becomes a source of nitrate during the downwelling periods. The seasonal cycle of the chlorophyll-a (CHLa) concentration follows that of nitrate, confirming that the primary production in this region is mainly driven by nitrate availability. The study also highlights the role of the Angola Current in transporting low-nitrate waters from the Equatorial Undercurrent, which influences the nitrate and CHLa balance in the Congolese upwelling system. These findings provide new insights into the mechanisms governing nutrient dynamics and biological productivity in the Congolese upwelling system. Understanding these processes is crucial for assessing the impact of climate variability on the regional marine ecosystems and fisheries.

Keywords: Nitrate Budget, Congolese Upwelling System, NEMO-PISCES Model, Physical-

Biogeochemical Interactions, Seasonal Variability, Coastal Trapped Waves

97

#### 1 Introduction

The Eastern Boundary Upwelling Systems (EBUS) are the most productive areas in the global ocean 103 in terms of biological resources, hosting almost 20% of the world's fisheries (Chavez and Messie, 2009, 104 2014), even though they only represent around 1% of the world's ocean surface (Freon et al., 2009). They are therefore an important economic support for the countries bordering these areas. EBUS are 105 controlled by wind stress blowing parallel to the coast, generating an offshore Ekman transport leading 106 107 to coastal upwelling of cold and nutrient-rich waters, which trigger primary production in the euphotic 108 layer, with increased surface chlorophyll-a (CHLa) concentration visible on satellite images (e.g. 109 Gutknecht et al., 2013). Besides their ecological richness, EBUS have been recognized as significant 110 sources of greenhouse gases (CO2, N2O, CH4), due to the oxygen minimum zone developed in such 111 productive areas (Gutknecht et 2013; Bachèlery et al., 2016), which are key drivers of climate 112 variability, hence studying these areas is a priority for climate evolution. 113 A part from these EBUS, recent studies (Bachèlery et al., 2016, Kopte et al., 2017, Awo et al., 2022) 114 have shown that the variability of SST (Sea Surface Temperature, a key upwelling indicator) at the 115 eastern boundary of the Atlantic is not only impacted by local wind forcing, but also by remote forcing 116 initiated by the equatorial dynamics. Indeed, the equatorial Kelvin waves (EKW) propagating along the 117 equator and later poleward along the coast as coastal trapped waves (CTW), can lead to the establishment 118 of seasonal upwelling systems. 119 This is the case for the Congolese and Angolan tropical upwelling systems, located in the south-east of the Gulf of Guinea (GG), which are highly productive marine ecosystems (Ostrowski et al., 2009). 120 Fishing provides around 25% of the Angolan population's total animal protein intake and is essential for 121 economic security (Hutchings et al., 2009; Sowman and Cardoso, 2010; FAO, 2022). The seasonal 122 variability of SST along the Angolan coasts shows an evolution that is similar to that observed in the 123 124 Congo (Bachèlery et al., 2015, 2016; Kopte et al., 2017; Awo et al., 2022; Brandt et al., 2023). It is characterized by a semi-annual pattern with an initial warming in February-April followed by a first 125 126 upwelling-induced cooling in May-August. Then there is a second warming, less significant than the 127 first, in September-November, followed by a second, less significant cooling in December-January. According to Radenac et al. (2020) and Brandt et al. (2023), in the tropical Atlantic ocean, the 128 129 thermocline and nitracline are often found at the same depth, which means that an upward movement of 130 the thermocline is associated with upward advection of nitrate, fueling biological productivity. Brandt et al (2023) also points out that in the Angolan tropical upwelling system, the seasonal cycle of nitrate 131 132 is in phase with that of CHLa derived from ocean color satellites (Fig.1). Given that the wind stress 133 along the coast is low for most of the year and out of phase with the upwelling period, the upwelling 134 would rather be induced by the passage of waves trapped at the coast, twhich signature is visible on the https://doi.org/10.5194/egusphere-2025-5112 Preprint. Discussion started: 11 November 2025 © Author(s) 2025. CC BY 4.0 License.

135 seasonal cycle of the sea level anomaly (Bachèlery et al., 2016; Awo et al., 2022; Brandt et al., 2023). 136 The cross-shore extension of the zone covered by upwelling is modulated by the zone's ocean circulation 137 (Fig. 1), dominated by the southward coastal Angola Current and Congo-Gabonese Undercurrent (Kopte et al., 2017; Bachèlery et al., 2016, Awo et al., 2022). Bachèlery et al (2016), based on a coupled 138 physical-biogeochemical model, shows that equatorial remote forcing is dominant for the interannual 139 140 variability of nutrients and primary production, whereas the local wind stress forcing is dominant for 141 the sub-seasonal variability. Some studies also highlight the important role played by turbulent mixing, locally enhanced in shallow waters near the coast (Körner et al., 2023; 2024; Tchipalanga et al., 2018a; 142 143 Rouault., 2012), in the seasonal modulation of SST and nutrients in Angolan coastal waters. 144 The Gabon-Congo upwelling zone (from 0°N to 6°S) is poorly documented. In a recent study, using a 145 high resolution (1/36°) simulation of NEMO model over the GG, Ngakala et al. (2025) assess the 146 seasonal mixed layer heat budget in the Congolese upwelling system. They found that the mixed layer 147 heat budget in the Congolese coastal area was driven by two major processes, warming by the heat 148 fluxes, dominated all year long by the solar flux, and cooling by the vertical mixing at the base of the mixed layer. Whereas the total advection contribution is less important and plays a secondary role in the 149 mixed layer heat budget. They have also mentioned that the relative contribution of vertical advection 150 and diffusion in the mixed layer heat budget is sensitive to the criterion used to define the mixed layer. 151 152 These results are in agreement with the conclusion of Körner (2023) in the northern Angolan upwelling south of the Congo river mouth. This later study found that the net surface heat flux warms the coastal 153 154 water further, whereas turbulent mixing across the base of the mixed layer is an important cooling term. 155 Also Scannell and McPhaden (2018), using data from a PIRATA mooring located off the Congo River at [8°E; 6°S], found that the seasonal evolution of mixed layer properties has two main phases: a warm-156 157 fresh phase (December-April) when solar heating is very efficient in warming SST in a thin mixed layer. 158 A cold-salty phase (May-September) driven by intensification of southeasterly trades in response to the 159 onset of the west african Monsoon and northward displacement of ITCZ. They also point out the necessity to take into account precipitation influence in the mixed layer heat budget. 160 161 Little information is available at seasonal scale for biogeochemistry in the GG. In the equatorial upwelling system, the respective contributions of physical and biological processes on the seasonality 162 163 of nitrate and biological productivity, in the mixed layer and euphotic layer, have been quantified by 164 Radenac et al (2020). Along the coast, most studies of biogeochemical dynamics are either limited to the section between 6°S and the Angola Benguela frontal zone (Brandt et al, 2023), or focus on the 165 interannual variability of biogeochemical tracers (Bachèlery et al, 2016). The seasonal variability of 166 biogeochemical tracers and biological productivity between 0°N and 6°S is likely influenced by three 167 168 major processes: coastal upwelling, input of nutrients by the Congo River discharge which is the second 169 largest river discharge in the world (Hopkins et al, 2013), and the stratification linked to the Congo River

freshwater. The aim of this paper is to highlight the respective roles of physical and biological processes in the seasonal cycle of nitrate and CHLa concentration and the respective contributions of coastal upwelling and Congo River discharge to the biological productivity in the Congolese upwelling system.

Figure 1: Average CHLa concentration in Gulf of Guinea with superimposed circulation pattern. The surface current (solid arrows) and thermocline current (dashed arrows) branches shown are the the Guinea Undercurrent (GUC); the Guinea Current (GC); the Equatorial Undercurrent (EUC); the northern, central and southern branches of the South Equatorial Current (nSEC, cSEC and sSEC); the South Equatorial Counter Current (SEUC); the South Equatorial Counter Current (SEUC); the Gabon-Congo Undersea Current (GCUC) and the Angola Current (AC). Also shown are Niger, Congo, Kouilou, Sanaga, Cavally, Volta, Ougooué, Cuanza and Kunene rivers. The red box indicate the coastal extent of the Congolese Upwelling System (CUS; 3–6° S, 1° wide coastal strip). CHLa concentrations data are derived from the CCI product for 2011.

### 2 Data and methods

# 2.1 Numerical model

To understand the dynamics in the Congolese upwelling system, we have used the NEMO (Nucleus for European Modelling of the Ocean) ocean general circulation model based on the primitive equations discretized on an Arakawa-C grid (Madec et al., 2024). The vertical mixing is computed from a turbulent closure scheme using the GLS (Generic Length Scale) formulation.

198

204205

208209

210211

218219

In this work, the NEMO model was coupled with PISCES (Pelagic Interactions Scheme for Carbon and Ecosystem Studies), a biogeochemical model developed by Aumont et al. (1998) and subsequently improved. Here, the version used is PISCES-2 (Aumont et al., 2015). This model has three main compartments: the first represents nutrients, including nitrogen compounds (nitrate and ammonium), iron, phosphate and silicates; the second represents phytoplankton and includes two classes, nano-phytoplankton and diatoms; the third compartment represents zooplankton, made up of two classes, microzooplankton and meso-zooplankton. We used the PISCES (cell quotas) model with constant Redfield ratios (Aumont et al., 2015).

A regional configuration of the GG (11°S - 6°N; 10°W - 14°E) with an horizontal resolution of 1/36° and 50 vertical levels is used. The atmospheric forcing is derived from the JRA-55 reanalysis of the Japanese meteorological agency (Kobayashi et al. 2015), except for the wind forcing is based on daily ASCAT (Advanced SCATterometer) satellite data at 1/4° spatial resolution. Lateral boundaries conditions are from Mercator GLORYS12V1 reanalysis data at 1/12° spatial resolution for physics and NEMO-PISCES reanalysis at 1/4° of Radenac et al (2020) for biogeochemistry. Continental freshwater inputs for this configuration are derived from the ISBA-CTRIP model, and in situ data from the HYBAM network for the Congo River. The NEMO configuration, ran over the period 2007-2017 (after a two-year spin-up), was validated by Ngakala et al. (2025) in our region of study. This simulation has also been validated and used in the Northern Gulf of Guinea for the coastal upwelling in summer and its interaction with mesoscale dynamics (Thiam et al., 2024). The reference simulation of the coupled biogeochemical physical model (NEMO-PISCES) was produced over the period 2007-2011, with a spin-up of 4 years for the biogeochemical part (2007-2010). We analysed monthly and daily outputs for the year 2011 monthly and daily outputs.

#### 2.2 Satellite and in-situ data

Several observational products were used to assess the model's ability to reproduce the physical and biogeochemical characteristics of the area for the year 2011. We used the MUR product (Multi-scale Ultra-high Resolution; Chin et al., 2017) with 1/4° spatial resolution and daily temporal resolution to assess the regional distribution and the seasonal cycle of SST. The vertical temperature distribution was assessed using the World Ocean Atlas (WOA; Locarnini et al., 2018; Zweng et al., 2019) climatology. The **CHLa** data used came from the Globcolour (https://www.globcolour.info/CDR Docs/GlobCOLOUR PUG.pdf) distributed by Copernicus Marine Environment Monitoring Service (http://marine.copernicus.eu/), which combines data from four ocean color satellites, with very high spatial resolution (1 km) and daily temporal resolution.

The nutrient fields were assessed using the CSIRO Atlas of Regional Seas climatology (Dunn and Ridgway, 2002) which merges several in situ databases (Argo buoys, WOD2005, WOCE3, Global Hydrographic Program, CTD and CMAR4 hydrology archives, NIWA5 hydrographic data, and CRC6

- hydrographic data). It provides physical variables (Temperature, Salinity) and biogeochemical variables 224 (NO<sub>3</sub>, PO<sub>4</sub>, O<sub>2</sub>, Si) both at the surface and at depth with a horizontal resolution of 1/2 °, 79 vertical levels from the surface to 5500 m depth, with a step of 5 m near the surface then increasing with depth, and a 225 226 daily temporal resolution. This product was built from 2009 and contains data from 1940 until 2011 227 which was the date when the last revision of the product was made. Near surface currents from the Ocean Surface Current Analysis Real-time (OSCAR, Johnson et al. 2007) dataset are based on satellite 228 and in situ measurements of sea surface height, surface vector wind and SST. datasetThey are derived 229 230 from quasi-linear and steady flow momentum equations thus combine geostrophic, Ekman and Stommel 231 shear dynamics. based on satellite and in situ measurements of sea surface height, surface vector wind 232 and SST. OSCAR product is available on a 1/3°×1/3° grid with a temporal resolution of 5 days for year 233 2011, and we use it to validate the near surface currents (first 30 meters) of the model outputs. Sea Level 234 Anomaly was computed from the salto/duacs gridded product of Absolute Dynamic Topography for 235 2011. This product is based on sea surface height measurement of multimission altimeters since 1992, optimally interpolated onto 0.25° x 0.25° longitude/latitude grid (Ducet et al., 2000). 236
- **2.3 Methods**
- The variability of nutrients and in particular nitrate is driven by several physical and biogeochemical processes taken into account in our model. As in Radenac et al (2020), the nitrate budget integrated over
- the mixed layer depth is represented by the following equation:

$$241 \quad \frac{\partial < NO_3>}{\partial t} = - < u \frac{\partial NO_3}{\partial x}> - < v \frac{\partial NO_3}{\partial y}> - < w \frac{\partial NO_3}{\partial z}> + \frac{1}{h} \left(K_Z \frac{\partial NO_3}{\partial z}\right)_{z=-h} - \frac{1}{h} \frac{\partial h}{\partial t} \left(< NO_3> - < w \frac{\partial NO_3}{\partial z}> + \frac{1}{h} \left(K_Z \frac{\partial NO_3}{\partial z}\right)_{z=-h} - \frac{1}{h} \frac{\partial h}{\partial t} \left(< NO_3> - < w \frac{\partial NO_3}{\partial z}> + \frac{1}{h} \left(K_Z \frac{\partial NO_3}{\partial z}\right)_{z=-h} - \frac{1}{h} \frac{\partial h}{\partial t} \left(< NO_3> - < w \frac{\partial NO_3}{\partial z}> + \frac{1}{h} \left(K_Z \frac{\partial NO_3}{\partial z}\right)_{z=-h} - \frac{1}{h} \frac{\partial h}{\partial t} \left(< NO_3> - < w \frac{\partial NO_3}{\partial z}> + \frac{1}{h} \left(K_Z \frac{\partial NO_3}{\partial z}\right)_{z=-h} - \frac{1}{h} \frac{\partial h}{\partial t} \left(< NO_3> - < w \frac{\partial NO_3}{\partial z}> + \frac{1}{h} \left(K_Z \frac{\partial NO_3}{\partial z}\right)_{z=-h} - \frac{1}{h} \frac{\partial h}{\partial t} \left(< NO_3> - < w \frac{\partial NO_3}{\partial z}> + \frac{1}{h} \left(K_Z \frac{\partial NO_3}{\partial z}\right)_{z=-h} - \frac{1}{h} \frac{\partial h}{\partial t} \left(< NO_3> - w \frac{\partial NO_3}{\partial z}> + \frac{1}{h} \left(K_Z \frac{\partial NO_3}{\partial z}\right)_{z=-h} - \frac{1}{h} \frac{\partial h}{\partial t} \left(< NO_3> - w \frac{\partial NO_3}{\partial z}> + \frac{1}{h} \frac{\partial NO_3}{\partial z}> + \frac{1}{h$$

$$-NO_{3z-h}$$
)  $+SMS(NO_3) >$  (1)

In this equation, the term on the left is the total nitrate tendency, where <> represents the vertical average in the mixed layer of depth h, which is defined as the depth where the potential density exceeds 244 245 the reference density, taken at 10 meters, by 0.03 kg/m3 (de Boyer-Montégut et al., 2004). On the right 246 of the equation, the first 3 terms are respectively the zonal, meridional and vertical advections of the nitrate, with u, v, and w the zonal, meridional and vertical components of the velocity field. The 4th term 247 is lateral diffusion. The 5th term is the vertical diffusion, where Kz is the vertical diffusion coefficient 248 that varies in space and time in the simulation. The 6th term corresponds to the entrainment at the base 249 250 of the mixed layer. Adding to these physical processes, the SMS term (last term on the right of the equation 1) is the source minus sink term, which takes into account the influence of biological processes 251 252 on the spatial and temporal variability of NO<sub>3</sub> concentrations. It is made up of several processes and is 253 represented by the expression:

$$SMS(NO_3) = Nitrif - \mu_{NO_3}^P * P - \mu_{NO_3}^D * D - R_{NH_4} * \lambda_{NH_4} * \Delta(O_2) * NH_4 - R_{NO_3} * Denit$$
 (2)

- where Nitrif corresponds to nitrification, which is the conversion of ammonium into nitrate by bacterial
- activity. It is parameterized by:

$$Nitrif = 1 - 1 + PAR > \lambda_{NH_4} - \frac{NH_4}{1 + \langle PAR \rangle} (1 - (O_2))$$
 (3)

- where NH<sub>4</sub> is the ammonium concentration, <PAR> is the average fraction of solar radiation available
- for photosynthesis,  $\lambda NH_4$  is the nitrification rate and (O2) is the oxygen variation in the mixed layer,
- which provides information on the oxic and anoxic conditions of the water column. The second and
- third terms on the right of equation (2) are the growth of nanophytoplankton and diatoms, where  $\mu_{NO3}^P$
- and  $\mu^D_{NO3}$  are their growth rates, P and D are their concentrations respectively.  $R_{NH4}$  and  $R_{NO3}$  are the
- stoichiometric N/C ratios of ammonification and nitrification respectively. Denit represents
- denitrification which occurs when the water becomes anoxic, and so nitrate (instead of oxygen) is
- consumed by remineralization of organic matter. A detailed description of the terms of these equations
- is given by Aumont et al (2015). We used the parameter PISCES values modified for the Tropical
- Atlantic ocean from Radenac et al. (2020). The balance terms in equation (1) have been calculated online
- for 2011. As the lateral diffusion term is generally negligible compared with the others, it will not be
- discussed further.
- Since the advection in (1) depends on both nitrate gradients and velocities, we investigate which
- component primarily controls its contribution. First, we analyze the seasonal evolution of gradient and
- velocity following Awo et al. (2022). Second, we evaluate the individual contributions of seasonal
- variations in velocity and gradient, as well as their combined effect, following Topé et al. (2023),
- according to equation (4).
- The Primary production (PP) was calculated from the phytoplankton evolution equation (Aumont et al
- 2015):

$$PP = \left(1 - \delta^P\right) \mu^P * P \tag{5}$$

- In this equation, P is the phytoplankton biomass (diatoms),  $\delta^P$  represents the exudation of the
- phytoplankton (diatoms).  $\mu^p$  is the specific growth rate of the phytoplankton taking into account nutrient
- and light availability. Note that this equation applies to each phytoplankton species (diatoms or
- nanophytoplankton), and total PP is the sum of PP from both diatoms and nanophytoplankton.  $\mu^p$  is the
- specific growth rate of the phytoplankton taking into account nutrient and light availability.
- Primary production, based on nutrients actually assimilated by phytoplankton and available light, is
- calculated online with the coupled NEMO-PISCES model. Primary production is the sum of new
- primary production (NPP) based on nutrients input by advection and/or diffusion and regenerated

- production (RPP) based on nutrients regenerated locally. NPP and RPP are also calculated online with
- the coupled model.

#### **3 RESULTS** 288

293 294

297

299

301

303

#### 3.1 Model/data comparison

#### 3.1.1 Spatial variations during the upwelling period

The assessment of our model simulation has been done using several observation products of physical variables (SST, SSH and currents) and biogeochemical tracers (NO3 and CHLa), based on both satellite 292 and in situ data. Figure 2 shows the regional distribution of observations and model outputs for both SST (Fig.2a-b), nitrate concentration (Fig.2c-d) and CHLa concentration (Fig.2e-f), averaged for austral 295 winter (June-August) which is the main Congolese upwelling period. As can be seen, the upwelling 296 feature is well captured by the model with cooling of surface water at the coast below 23°C in the model and 22°C in the MUR product. This cooling feature is consistent with high nitrate and CHLa 298 concentrations in both models and observation, particularly north of the Congo estuary (6°S) and nearby Kouilou River mouth at 4.47°S. These cool and enriched nutrient coastal waters are spread offshore 300 displaying a cross-shore gradient, with a greater extension in the observation than the model. The highest nitrate concentration in the coastal waters is greater than 10 mmolN.m<sup>-3</sup> in the model (8 mmolN.m<sup>-3</sup> in 302 the observation) located mainly in the Congo River plume area, inducing enhancement of PP resulting in a strong CHLa signature.

Figure 2: Comparison between model (left hand side) and observations (right hand side) with regional distribution of sea surface temperature (a, b), nitrate concentration (c, d) and CHLa concentration averaged for austral winter (June, July, August).

The offshore area (7°E-10°E) is the oligotrophic zone characterized by relatively warm waters (24.5°C in the model and 23.5°C in the observation), depleted in nitrate and less productive in CHLa concentration. In this offshore area nitrate concentrations are lower than 1.6 mmolN.m<sup>-3</sup> in the observation and 0.8 mmolN.m<sup>-3</sup> in the model.

Although the model captures relatively well the regional distribution of the 3 variables, we can see some differences. For instance, the model is warmer than observations by about 1°C and shows stronger nitrate concentration (by about 2 mmolN.m<sup>-3</sup>) and CHLa concentration (6-10 mg.m<sup>-3</sup>) at the coast. In the offshore area, the model seems to be less enriched in nitrate concentration than the observation by about 0.8 mmolN.m<sup>-3</sup>.

High variability of nitrate concentration is found in the coastal Congolese area (Fig.3) and in the Congo river plume zone as we can see in model annual standard deviation distribution of  $NO_3$  Therefore, the black box (3°S-6°S, 1° width coastal band) in Figure 3 is used to analyze the vertical nitrate profile to

322323

325

326327

329330

assess the model's ability to capture its vertical distribution. This area corresponds to our studied area in the Congolese coastal upwelling zone.

Figure 3: Regional distribution of standard deviation of nitrate concentration of the model

**Figure 4**: Comparison between model and observation using vertical distribution of nitrate concentration in the first 90 m, in the coastal box (3°S-6°S and 1° width band to the coast) in the upwelling season. black represents nitrate concentration isolines.

Very close to the surface, water masses are nutrient depleted for both model and observation (Fig.4), likely due to photosynthesis activity of phytoplankton that consumes nitrate in presence of light, increasing its biomass thus CHLa concentrations. However this depletion is more pronounced in the model than in the observation. In the subsurface, the high nitrate concentration is due to the

337338

340341

remineralization of organic matter by bacteria and coastal upwelling of deeper enriched nitrate waters, with the model showing higher concentrations than observed. Although nitrate isolines are shallower in the model than in observations below about 15 m depth, some nitrate isolines are relatively well captured by the model, for instance isolines 7 and 10 mmolN.m<sup>-3</sup>.

## 3.1.2 Seasonal cycle of SST, nitrate, SLA and current in the Congolese coastal area

Figure 5: Comparison between modeled (left) and observed (right) seasonal cycles of SST (a, b), nitrate concentration (c, d) and Sea Level Anomaly (e, f) averaged in the coastal box (6°S-3°S, 1° width).

Now we use the coastal box defined in Figure 4 to evaluate the seasonal cycle of nitrate. The nitrate variability is characterized by a semi-annual cycle with two maxima and two minima in the model and

342 the observations. The main maximum occurs from May to September when SST reaches its minimum 343 of 20°C in both model and observations (Fig.5c, Fig.5d) and the secondary maximum occurs in December when SST reaches a secondary minimum of 25.5°C in the model and 24.5°C in observation. 344 345 We have a warmer SST reaching 30°C from January to April and 26°C from October to November in both the model and the observations. This semi-annual SST cycle is likely due to CTWs propagation 346 since it is consistent with the SLA seasonal cycle (minimum SST corresponds to negative SLA and 347 348 maximum SST corresponds to positive SLA) as mentioned earlier by Ngakala et al. (2025) in the same area. Indeed, the propagation of CTWs induce vertical migration of the thermocline resulting thereby in 349 350 cooling or warming at the surface. At the seasonal scale, the propagation of upwelling CTWs from May to September and in November-December uplifts thermocline, supplying cold waters to the surface and 351 352 reducing Sea Surface Height (SSH) by steric effect. As downwelling CTWs propagate from January to 353 April and in September-October, they deepen thermocline, warming the surface and increasing SLA. 354 The cold waters upwelled by CTWs (May-September) are highly enriched in nitrate, whereas warm 355 surface waters induced by downwelling CTWs (January-April and September-October) are nitrate depleted. The seasonal variability of SLA due to CTWs (Fig.5a, Fig.5b) is consistent with the seasonal 356 357 variability of SST (Fig.5ca, Fig.5d) and of nitrate concentration (Fig.5e, Fig.5f) in both the model and observations. The highest nitrate concentration is around 10 mmol.m<sup>-3</sup> near the Congo River mouth 358 359 (6°S) and decreasing northward, however the observations seem to be richer in nitrate than the model. In December during the secondary cooling, nitrate concentration in the model is greater by about 1.2 360 mmol.m<sup>-3</sup> than observed along the coast. In the warming period (January-April and October-November), 361 this coastal area seems to be nitrate depleted. 362 However, despite this well captured SLA signature by the model, the seasonal cycle of SLA is more 363 364 intense in the observations than in the model. Previous studies (Bachèlery et al 2016, Kopte et al 2017, 365 Awo et al 2023, Brandt et al 2023) have shown that this sea level anomaly feature is due to combined 366 effect of Equatorial Kelvin Waves (EKW) remotely forced at the equator and the CTWs propagating poleward along the African coast. In summary, during the cooling period, the CTWs inducing upwelling 367 travel along the coast decreases SLA and raises in the thermocline which can be taken as nitracline proxy 368 (Radenac et al 2020), thereby enhancing nitrate supply at the surface. 369 370 High nitrate concentrations support biological production, therefore correspond to high CHLa signals at 371 the surface (Fig.6). On the contrary, during the warming period, the downwelling CTWs downwelling propagating along the coast increase SLA, lower the thermocline (Ngakala et al. (2025). This also lowers 372 373 the nitracline and consequently deplete the nitrate concentration at the ocean surface, thus the low CHLa 374 signal (Fig.6). 375 The variability of simulated near-surface currents between 0 to 15 m depth (Fig.7) was compared to the 376 OSCAR product. Here, we make a latitudinal section at 4°S and look at the seasonal cycle of meridional 377 currents (Fig.7a and 7b) from 7°E to the coast for both the model (Fig.7a) and observations (Fig.7b).

We do not restrict to the Congolese box as we have done for other variables, because the OSCAR product is not well resolved at the vicinity of the coast. So we can see that the model reasonably represents the seasonal variability of meridional currents with northward velocities in April, June-September and November-December with the magnitude of around 0.1 m.s<sup>-1</sup>. In the observations, this structure is more or less similar, but we can see some differences: southward velocities between 8.5°E and 10.2°E in July, and also between 7°E and 9°E during August-September and January, are not found in the model.

**Figure 6**: Seasonal cycle of CHLa concentration (in background) and wind stress (black arrow) along the Congolese coastal box (6°S-3°S, 1° width) area in the model.

Nevertheless, we can see in both products southward currents in February-March and October with a strong magnitude of 0.25 m.s<sup>-1</sup> in the observations though only 0.15 m.s<sup>-1</sup> in the model. This seasonal structure is consistent with the seasonal cycle of meridional currents off Angola, further south, and the southward flow in February-March and October seems to be the Angola current (Kopte et al 2017). To assess the zonal current (Fig.7c and 7d), we make a longitudinal section at 10°E and we look at the seasonal cycle of zonal current between 3°S and 7°S. The modeled zonal structure with westward velocities from April to August and November – December has a magnitude of about 0.1 m.s<sup>-1</sup> along the section (Fig.7c), in agreement with the observations (Fig.7d), which also show westward velocities from April to August. However there are some differences with the model from March to August between 3°S and 4.5°S where we can observe very strong (more than 0.5 m.s<sup>-1</sup>) eastward flow. In contrast to the model, the flow during November is eastward in the observation. The noticeable similitude is the eastward flow in January- February and October with a magnitude of 0.15 m.s<sup>-1</sup> which seems to be the signature of the South Equatorial UnderCurrent (SEUC). This eastward current is deeper further offshore (100 m depth) west of 0°E (Bourles et al., 2004) and rising near the surface near the coast (Nubi et al., 2016; Assene et al., 2022).

# 3.2 Nitrate budget in the mixed layer

Generally, the seasonal variations in CHLa are thought to be primarily related to seasonal variations of the nitrate input in the equatorial Atlantic ocean (Loukos and Mémery., 1999; Radenac et al, 2020) and in the tropical Angolan upwelling (Brandt et al., 2023). This is probably the case also in the Congolese coastal area, where the seasonal cycles of nitrate (Fig.8a) and CHLa-a (Fig.6) in our model are very consistent. Like the seasonal cycle of nitrate, the seasonal change rate of nitrate concentration (Fig.8b) has a semi-annual cycle with four phases: a first increasing phase between March and August with a highest amplitude (0.1 mmol.m<sup>-3</sup>.d<sup>-1</sup>) in July, followed by a decreasing phase in September –November with a highest amplitude in October (0.08 mmol.m<sup>-3</sup>.d<sup>-1</sup> north of 5°S and about 0.15 mmol.m<sup>-3</sup>.d<sup>-1</sup> between 5°S and 6°S).

**Figure 7**: Seasonal cycle of surface current, zonal current at 10°E section and between 3°S and 7°S and meridional current at 4°S section from 7°E to the coast.

Then we have a weak second increasing phase in November-December of about 0.03 mmol.m<sup>-3</sup>.d<sup>-1</sup> and a weak decreasing phase in January- February. This semi-annual cycle is due to a balance between nitrate supply by physical processes (Fig.8c), maximum during the main upwelling period, and nitrate consumption by biological processes (Fig.8d).

## 3.2.1 Seasonal Nitrate Budget Analysis: Horizontal Vs Vertical Contributions

Looking at our previous results, we saw that physical processes drive the nitrate supply in the Congolese upwelling system, now we will look at the contribution of horizontal and vertical processes to understand which are the main drivers for this nitrate supply. Figure 9 shows that horizontal and vertical processes (Fig.9a and Fig.9b respectively) are of great importance for nitrate supply.

**Figure 8**: Latitude-time Hovmöller diagram of the model seasonal cycle of Mixed Layer Nitrate (MLN) budget, a), the rate of the MLN change (b), the physical process contribution (c) and the biological process contribution (d) along the Congolese coast. Units are mmolN.m<sup>-3</sup> and and mmolN.m<sup>-3</sup>.d<sup>-1</sup> for Figure 7a and Figure 7b,c,d, respectively.

In fact, as we can see in the figure 8, vertical processes (Fig.9b) are the main driver of nitrate supply during the upwelling between 3°S and 5.5°S with an input magnitude of around 0.6 mmol.m<sup>-3</sup>.d<sup>-1</sup> along the coast while the horizontal processes are the main driver at the vicinity of Congo river mouth (5.5°S – 6°S) all year long. The latter seems to be dominated by zonal advection (fig.9c) with a very high nitrate input of more than 0.8 mmolN.m<sup>-3</sup>.d<sup>-1</sup> nearby 6°S with a northward extension, largest firstly in November-December and secondly in June-July. This is consistent with the seasonal maximum of Congo River discharge, which suggests a nitrate input through the river plume (Hopkins et al 2013). North of 6°S, meridional advection (Fig.9e) drives horizontal processes.

The nitrate budget analysis reveals also that vertical processes (Fig.9b) are dominated by vertical advection (Fig.9d), while vertical mixing has the same seasonality but a smaller contribution (Fig.9f). Indeed the nitrate input by the vertical advection is about 0.45 mmolN.m<sup>-3</sup>.d<sup>-1</sup> but only around 0.25

mmolN.m<sup>-3</sup>.d<sup>-1</sup> by vertical diffusion. Both vertical processes decrease nitrate concentration nearby 6°S, under Congo River plume influence with the dominant zonal advection contribution. Indeed, as nitrate concentration is greater in the near-surface Congo River plume than in subsurface (between 5 to 10 m, see Fig.4), deeper waters rising at the surface by vertical advection reduce nitrate in the plume area.

**Figure 9**: Latitude-time Hovmöller diagram of the model seasonal cycle of horizontal (a)and vertical (b) process contributions, zonal (c), meridional(e), vertical (d) advections, vertical diffusion (f) averaged in the mixed layer along the Congolese coast. Units are mmolN.m<sup>-3</sup>.d<sup>-1</sup>.

Similarly, vertical mixing of subsurface waters with the plume waters decreases nitrate concentration at the surface, although the strong haline stratification associated with the Congo River plume limits this effect.

454

463

468

470

It is important to note that, on the one hand, vertical advection and vertical diffusion have the same seasonality as SLA, in opposite phase, northward of 5.5°S. This suggests that the upwelling associated with CTWs (negative SLA) induce these vertical processes and therefore drive the input of nitrate in the northern part of the Congolese coast. On the other hand, horizontal advection (both zonal and meridional advections) has the same seasonality as the Congo River discharge between 5.5°S and 6°S. This suggests that the Congo River supplies nitrate through zonal advection, near its mouth.

Figure 10: Spatial distribution averaged over the main upwelling period of (a) nitrate, (b) nitrate tendency 459 contributed by (c) physical processes and (d) biological processes, all averaged in the mixed layer in Austral winter. The mean current in the mixed layer is superimposed in (a). Nitrate concentration units are mmolN.m<sup>-3</sup> and the tendency terms units are mmolN.m<sup>-3</sup>.d<sup>-1</sup>.

# 3.2.2 Regional Nitrate Budget Analysis in the main upwelling period: Physical Vs **Biological Contributions**

The regional distribution of nitrate balance terms averaged over the mixed layer for the austral winter (June-July-August), when upwelling reaches its maximum intensity, is presented in Figure 10. The nitrate tendency (Fig.10b) shows that during the upwelling period, nitrate input occurs throughout the domain with a high value of 0.08 mmolN.m<sup>-3</sup>.d<sup>-1</sup> in the plume zone and 0.06 mmolN.m<sup>-3</sup>.d<sup>-1</sup> along the coast in the northern part. In the offshore zone, the nitrate tendency is lower, with a magnitude of around 0.02 mmolN.m<sup>-3</sup>.d<sup>-1</sup>. This distribution of the nitrate tendency shows that the input of nitrate by physical processes (Fig.10c) is slightly greater than the uptake of nitrate by biological processes (Fig.10d) throughout the area, explaining the positive nitrate change rate. Figure 10a shows that nitrate inputs along the coast are very high over a width of about 1.5° longitude from the coast. During this main

490

493

496

501

507

period of upwelling, vertical processes (Fig.11b) dominate the physical contribution to nitrate supply 474 throughout the domain, except in the fluvial plume region where horizontal processes (Fig.11a) 475 dominate. Vertical advection dominates the nitrate supply along the coast and north of the river mouth 476 (Fig. 11d), the while vertical diffusion (Fig. 11f) appears to be the main driving process for the relatively 477 weak nitrate supply south of the river mouth in the northern part of the Angolan coast (between 6°S and 478 7°S).

## 3.3 Nitrate budget in the euphotic layer and along the water column

Now, in addition to the process acting in the surface mixed layer, we look for other processes involved in the nitrate budget below the mixed layer by analysing the nitrate budget in the euphotic layer, generally defined as the zone where light penetration exceeds 1% of the surface light, allowing for the presence of phytoplankton and other photosynthetic organisms. The seasonal variations of thermocline, mixed layer and euphotic layer depths are compared in Figure 12. In Fig.12a, the thermocline (20°C isotherm) variation is very close to the nitracline as suggested by Radenac et al (2020). The mixed layer euphotic layer is very shallow (~ 10 m) throughout the year probably due to the Congo River plume stratification. The euphotic layer extends deeper than the mixed layer, but is generally shallower than the thermocline, except from June to September. The euphotic layer gets thinner during upwelling through the enhancement of CHLa concentration which reduces light penetration (self-shadowing by CHLa). Overall the nitrate tendency (Fig. 12b) has the same semi-annual variation in the euphotic layer than in the mixed layer, although more intense in the euphotic layer with a maximum at the bottom of the euphotic layer. In fact, most nitrate input by physical processes (Fig. 12c) happens in the mixed layer, where it is almost balanced by biological nitrate uptake (Fig.12d). In contrast, in the euphotic layer below the mixed layer (between 10m and 40m depth), biological processes are poorly active and the nitrate variability is almost exclusively induced by physical processes. The mean nitrate input in the euphotic layer is about 0.1 mmolN.m<sup>-3</sup>.d<sup>-1</sup> during the main upwelling period and the maximum input (0.2 mmolN.m<sup>-3</sup>.d<sup>-1</sup>) occurs in May at the base of the euphotic layer (Fig. 12b).

In the euphotic layer, biological activity is dominated by photosynthesis which removes nitrate, whereas 499 below the euphotic layer remineralization supplies nitrate with about 0.05 mmolN.m<sup>-3</sup>.d<sup>-1</sup> almost all year 500 long (Fig. 12d).

#### 3.3.1 Euphotic Layer Nitrate Budget Analysis: Horizontal Vs Vertical Contributions

Figure 13 shows that, even in the euphotic layer, the physical contribution to nitrate supply (Fig.12c) is mostly driven by vertical processes (Fig.13b). However, the large decrease in nitrate (Fig.12b) in 503 October is also caused by horizontal contributions (Fig. 13a) in the euphotic layer. Between the surface 504 505 and 5m depth, horizontal processes dominate nitrate input (Fig.13e). In the euphotic layer below, 506 meridional advection is the main driver of nitrate removal almost year-round, particularly in June and October.

**Figure 11**: Contribution of (a) the horizontal processes, including (c) zonal advection and (e) meridional advection, and (b) vertical processes, including (d) vertical advection and (f) vertical diffusion, to the nitrate budget averaged in the mixed layer during the austral winter. Units are mmolN.m<sup>-3</sup>.d<sup>-1</sup>.

Zonal advection (Fig.13c) supplies nitrate in the euphotic layer, with a maximum above the mixed layer depth, throughout the year. This nitrate input is more than compensated by nitrate lost by meridional contribution, below the upper 5 m (Fig.13e), except in June-July-August. Vertical advection (Fig.13d) is the dominant vertical process (Fig.13b) in the nitrate budget. Below 30 m depth, it has a semi-annual cycle characteristic of upwelling and downwelling CTWs propagation, associated with nitrate increase when the thermocline shallows and nitrate decrease when the thermocline deepens, with the maximum and minimum values around the thermocline depth. Moreover, vertical advection supplies more nitrate in the mixed layer than in the euphotic layer below during the main upwelling period (June, July and August), but rather the opposite during the second upwelling period (December). Between the mixed layer depth and the euphotic layer depth, vertical diffusion (Fig.13f) tends to partially compensate for the effects of vertical advection on nitrate. However, in the mixed layer it mostly supplies nitrate, particularly during the upwelling seasons.

526527

532533

534535

**Figure 12**: Depth-time Hovmöller diagram of the model seasonal cycle of nitrate concentration (a), nitrate tendency in the euphotic layer (b) contributed by physical process (c) and biological process (d) in the congolese coast coastal box (Fig. 3). The units are in mmolN.m<sup>-3</sup>.d<sup>-1</sup> for all plots (b, c, d) except nitrate concentration (a) which is in mmolN.m<sup>-3</sup>. The black line represents the thermocline. The dashed magenta and black lines are the euphotic layer and mixed layer depths respectively.

## 3.3.2 Nitrate budget analysis: advection components analysis

As nitrate advection depends on velocity and on the nitrate gradient, we now evaluate the individual contributions of seasonal variations in velocity and nitrate gradient, as well as their combined effect, to the seasonal variations of nitrate advection (see section 2.3, equation 4).

# 3.3.2.1 Nitrate budget analysis: horizontal advection

Figure 14 allows to visually compare the depth-time structure of the zonal nitrate advection (Fig.14a) with that of its different components, while correlation r is used to quantify the comparison.

**Figure 13**: Depth-time Hovmöller diagram of the model seasonal cycle of contributions to the nitrate budget of horizontal and vertical processes (a and b respectively), zonal, meridional, vertical advections (c, e and d respectively), vertical diffusion (f) along the Congolese coast (3°S-6°S and 1° width to the coast). Units are mmolN.m<sup>-3</sup> for all of the plots. The strong black line represents seasonal variation of the thermocline, the dashed magenta line is the euphotic layer and dashed black line is the mixed layer depth.

The zonal nitrate advection seasonal cycle in the 0-100 m water column (Fig. 14a) is controlled first by the term  $u > \Delta < \frac{dNO3}{dx} >$  (Fig.14e, r=0,77), i.e. the annual mean zonal current multiplied by the seasonal variations of the nitrate zonal gradient, second by the term  $\Delta < u > < \frac{dNO3}{dx} >$ , (Fig.14d, r=0,49), i.e. the seasonal variations of the zonal current multiplied by the annual mean nitrate zonal gradient, and third (and much less) by the term. The third component, which represents the simultaneous variation in zonal current and nitrate gradient ( $\Delta < u > \Delta < \frac{dNO3}{dx} >$ , (Fig.14f, r=-0,15), i.e. the product of seasonal variations of both the zonal current and the nitrate zonal gradient. The seasonality of the zonal current (Fig. 14c) is influenced by the seasonal cycle of the South Equatorial Undercurrent SEUC, with maximum values in September-October and February-March (Dorothée et al, 2004) in this zone (3°S-6°S, 1° from the coast). Thus we can conclude that the SEUC plays a key role

in the nitrate balance in the Congolese system by bringing nitrate in February-March and in September-October to the euphotic layer.

Figure 14: Depth-time Hovmöller diagram of the model seasonal cycle of nitrate advection (a), nitrate gradient (b), current (c), current variation times mean nitrate gradient (d), gradient variation times mean current (e) and gradient variation times current variation (f), all along the zonal axis in the congolese coastal box of Fig. 3. Units are milli mol per cubic meter per day for all of the plots except (b) (milli mol per cubic meter) and (c) (meter per second). The full black line represents the thermocline, the dashed magenta line is the euphotic layer depth and the dashed black line is the mixed layer depth.

In the euphotic layer, we see that, as for the meridional nitrate advection, the meridional current (Fig.15c) looking at the shape seems to be the main factor in the vertical and temporal variation shape of the meridional nitrate advection (Fig.15a). The Angola current (AC), is the factor which modulates nitrate lost by meridional nitrate advection throughout the year, with maximum loss in September-October and June-July, in and below the euphotic layer, except In the first 5 m-depth.

Our analysis in this section reveals that the simultaneous variation in both the current and the gradient  $(\Delta < v > .\Delta < \frac{dNO3}{dy} >$ , Fig.15f) exhibits the highest correlation (52.7%) with meridional nitrate advection, thus explaining the major changes in advection. This result highlights the significant impact of the concurrent variability of both current and gradient on meridional nitrate advection in the euphotic layer. In contrast, gradient variation ( $v > .\Delta < \frac{dNO3}{dy} >$ , Fig.15e) is poorly correlated (9.7%) with meridional nitrate advection. The very low positive correlation with total meridional advection (a) indicates that this term, representing the effect of a mean current acting on a fluctuating nitrate gradient. This suggests that either the mean meridional current is weak, or its interaction with the fluctuating

576577

578579

587

592

596

599 gradient does not lead to significant changes in overall advection. The variation in the current ( $\Delta < v > ... < \frac{dNO3}{dy} >$ , Fig.15d) shows the lowest correlation (-28.7%). This low, negative correlation with total meridional advection (Fig.15a) indicates that this term, representing the effect of fluctuating currents on a relatively stable mean nitrate gradient, is *not* a dominant driver of the overall meridional nitrate advection. In fact, a negative correlation suggests it might weakly oppose the main advection pattern or have an inverse relationship. This implies that the mean gradient is either small or the current variations are not aligned to produce significant advection changes via this mechanism.

#### 3.3.2.2 Nitrate budget analysis: vertical processes

Our analysis reveals that within the euphotic layer, the variation in the vertical gradient ( $< w > . \Delta <$  $\frac{dNO3}{dz}$  >, Fig.16e) appears to better explain the variation in vertical nitrate advection (Fig.16a), showing a correlation of 79%. In contrast, vertical velocity variation ( $\Delta < w > . < \frac{dNO3}{dz} >$ , Fig.16d) plays a secondary role, with a correlation of approximately 64.6% with vertical nitrate advection. Looking now at the vertical nitrate advection, we can see strong similarities in the vertical and temporal variation structure between vertical advection seasonality (Fig. 16a), vertical nitrate gradient seasonality (Fig. 16b) and vertical velocity seasonality (Fig. 16c), both three are very strong in the euphotic layer. We can also see in the semi-annual vertical velocity that from April to August and November-December, vertical velocities are upward corresponding to negative values of SLA (Fig.5a, 5b), lowest SST (Fig.5c, 5d) values and highest nitrate concentration (Fig.5e, 5f). From January to March and September-October, vertical velocities are downward corresponding to positive values of SLA (Fig.5a, 5b), highest SST (Fig.5c, 5d) values and lowest nitrate concentration (Fig.5e, 5f). This later observation confirms that CTWs propagating from April to August and November-December are associated with upwelling. In contrast, CTWs propagating from January to March and September-October are associated with downwelling. Note that similar results were found by Ngakala et al. (2025) for the seasonal heat budget in the Congolese upwelling (from 4°S-6°S and 1° width to the coast) and also further south in the Angolan upwelling by Korner et al. (2024).

Figure 15: Depth-time Hovmöller diagram of the model seasonal cycle of nitrate advection (a), nitrate gradient (b), current (c), current variation times mean nitrate gradient (d), gradient variation times mean current (e) and gradient variation times current variation (f), all along the meridional axis in the congolese coastal box of Fig. 3.

Units are milli mol per cubic meter per day for all of the plots except (b) (milli mol per cubic meter) and (c) (meter per second). The full black line represents the thermocline, the dashed magenta line is the euphotic layer depth and the dashed black line is the mixed layer depth.

However, if we consider the first hundred meters, vertical velocity variation has the highest correlation of 83.1% with vertical advection, whereas vertical gradient variation has only 63% of correlation with vertical advection. The third component ( $\Delta < w > .\Delta < \frac{dNO3}{dz} >$ , Fig.16f) has a lower negative correlation -6.2% with vertical advection. During the upwelling period, the variation in vertical gradient (Fig.16e) at the base of the mixed layer has a much greater influence on nitrate supply than the variation in vertical velocity (Fig.16e). Another observation is that in the same main upwelling period there is a lag between maximum vertical current which happens in May and the maximum vertical gradient indicated by the shallowest thermocline in July. This lag results in a highest input of nitrate by vertical advection (Fig.16a) in the mixed layer in June. It can be seen that the nitrate output by vertical advection during the downwelling period is mostly induced by the vertical nitrate gradient in the mixed layer whereas deeper in the euphotic layer these losses are induced by vertical downward velocities induced by downwelling CTWs propagation.

#### 4 Discussion

In this section, we discuss our results: the model-data comparison, the influence of the mixed layer criteria, the main nitrate budget driver in the Congolese upwelling systems compared to the other tropical Atlantic upwelling systems. Finally, we will explore the factors governing seasonal productivity in the

 Congolese upwelling system, integrating our understanding of physical forcing and nutrient availability to characterize its biological response. Through this comprehensive discussion, we aim to provide a nuanced understanding of the oceanographic processes at play in the coastal Congo region and the capabilities and limitations of our modeling approach.

Throughout this work, we have shown that our model reasonably reproduces the observations in terms of temperature, nitrate, CHLa, SLA and surface current, although there are a few differences that we will discuss in this section. First of all, we saw that temperature in our model is greater than observed by around 1°C in regional distribution as well as in seasonal cycles. This is a common bias in ocean and climate models in the Eastern tropical Atlantic. Indeed, several studies suggest that this warm bias is due to models' deficiency in simulating low-level clouds, resulting in overestimation of shortwave radiation (e.g. Xu et al., 2014).

Figure 16: Depth-time Hovmöller diagram of the model seasonal cycle of nitrate advection (a), nitrate gradient (b), current (c), current variation times mean nitrate gradient (d), gradient variation times mean current (e) and gradient variation times current variation (f), all along the vertical axis in the congolese coastal box of Fig. 3.

Units are milli mol per cubic meter per day for all of the plots except (b) (milli mol per cubic meter) and (c) (meter per second). The full black line represents the thermocline, the dashed magenta line is the euphotic layer depth and the dashed black line is the mixed layer depth

With regard to nitrate concentrations, the regional distribution shows that north of the mouth of the Congo River and near the coast, the model agrees well with CARS climatology. But offshore and south of the mouth of the Congo River, the model underestimates nitrate concentrations. Similarly, in the seasonal cycle, we see that the model captures the seasonal variability, but underestimates the amplitude compared to the data. These biases may be explained by the temporal coverage of the CARS

649

653654

656

661

662663

668

672673

676

679

681

climatology, which covers a long period (from 1940 to 2011) of data (Bachèlery et al., 2016) compared to our model which covers only one year (2011). Another bias may be the lack of data in CARS, in our study area. We did not show the comparison of our model with the WOA climatology due to its low resolution (1°) for nutrients and oxygen in our coastal studied area compared to CARS climatology (0.5°). However the biases with WOA are similar as with CARS even if WOA covers a more recent period (from 2005 to 2017). The data coverage and individuals data profiles are also available for WOA, confirming that very few data are available in our studied area. The differences in the surface CHLa concentration between the model and the satellite observations may be associated with a lack of data for the ocean colour satellite observations, particularly in August, due to the cloud cover which induces atmospheric contaminations (Hardman-Mountford and McGlade., 2002; Estival et al., 2013) of the satellite signal, resulting in a lack of CHLa signal (Nieto et al., 2016). The biases in the surface currents between the model and the observations are mainly due to the under-estimation of the currents in the OSCAR product near the coastal zone (Sikhakolli et al., 2013), especially in our studied area where very few data are available. Despite these results discussed earlier in this paper, we have to keep in mind that there are also some limitations in our simulation. For example, the model has difficulties reproducing the seasonal cycle of CHLa concentration, in particular the first CHLa blooms occurring in February-March, highlighted by the MODIS ocean colour satellite data. This might be due to the CHLa concentration of the Congo river in our model. Indeed, we take into account the nutrients and dissolved organic matter discharges of the river but not the CHLa concentrations (information not available in the HYBAM database). We can also observe a minimal or slightly reduced concentration of CHLa around the mouth of the Congo River in our model (Fig. 2e). This is explained by the very high speed (greater than 2 m/s) of the Congo River current at its mouth in our model. This has resulted in the transport of chlorophyll produced by phytoplankton photosynthesis away from the mouth of the Congo River. Our analysis shows that, in the Congolese upwelling system, the nitrate budget in the mixed layer is dominated by the physical processes during the upwelling period, particularly vertical advection, while zonal advection and vertical mixing play a secondary role. In contrast to these latter results, Ngakala et al. (2025) using a high resolution simulation (1/36°) of NEMO in the Congolese upwelling, have shown through a mixed layer heat Budget, that vertical diffusion was the main contributor of cooling during upwelling period in the mixed layer. They found that the vertical advection has a secondary role in cooling of the mixed layer. These differences can be associated with the definition criterion of the mixed layer depth, which is very shallow in their analysis (0.2 °C temperature criterion from 0.5 m depth as reference depth of density variation). They state that, if defining instead the mixed layer depth with the de Boyer-Montégut criterion, as we do, then vertical advection would play a greater role than vertical diffusion, as we find in our nitrate budget. Thus, as mentioned in our study and in agreement with Ngakala et al. (2025), the relative contributions of vertical advection and diffusion depend on the definition of the mixed layer depth.

685

688

690

697

698699

711

714715

concentration of nitrate, as in other tropical Atlantic upwelling systems (Radenac et al., 2020). However different processes drive the seasonal cycle of nitrate and CHLa in the different tropical Atlantic upwelling systems. In the Equatorial Atlantic upwelling system, the seasonal cycle of nitrate and CHLa are driven by the wind stress and wind stress curl (Caniaux et al., 2011, Radenac et al., 2020). In the Tropical Angola Upwelling system, the main driver of these seasonal cycle are the CTWs as in our area with a main peak in austral winter (May-July) and a second peak in December-January. However vertical mixing plays also a key role in the Tropical Angola Upwelling system (Awo et al., 2022; Ostrowski et al., 2009; Körner et al., 2023) due to onshore propagating internal tide waves interacting with sloping topography (Brandt et al., 2023). In the upwelling systems of the equatorial Atlantic and tropical Angola, vertical mixing is the main driver of nitrate input to the mixed layer. This is due to local mechanisms that occur in these areas (local forcing), such as the intensification of the vertical shear stress between the South Equatorial Current (SEC) and the Equatorial Undercurrent (EUC) at the equator (Jouanno., 2010, Radenac et al., 2020) and the dissipation of internal tide that interact with the continental shelf and produce turbulent mixing at the Angolan coast (Körner et al., 2023, Zeng et al., 2021, Brandt et al.). In the Congolese system, we can suggest that the strong stratification induced by the discharge of the Congo River, which is the second largest river in the world, contributes to thinning the mixed layer, limiting the effect of mixing very close to the surface. In the euphotic layer and below, the nitrate budget is almostly dictated by physical processes, which are mainly modulated by currents that transport water of different properties. In this section, we will discuss the influence of currents on the nitrate and CHLa balance in the euphotic layer. We noted that vertical and zonal advections were the drivers of nitrate input in the upwelling period, while vertical mixing and meridional advection were the drivers of nitrate losses in the lower part of the euphotic layer (just below the mixed layer depth) in this period. However, during the downwelling period, vertical mixing mostly brings nitrate to the lower part of the euphotic layer, while vertical and meridional advection always remove nitrate. Meridional advection is therefore the main factor in nitrate loss in the euphotic layer all year long. This is consistent with the warming effect of meridional advection shown by Körner et al. (2023) in the Angolan upwelling. Radenac et al (2020) showed that in the equatorial euphotic layer, zonal advection by the EUC current was the main driver of nitrate losses, which may explain our previous results since the southward Angola current dominating the ocean circulation in the Congo-Angola zone is fed by the EUC current. Indeed, the EUC, whose source waters come from the oligotrophic layers of the subtropical South Atlantic, has relatively low nitrate concentrations compared to neighbouring waters (Schott et al., 1998; Johns et al., 2014; Tuchen et al., 2022a). These low nitrate waters are brought to the Congo-Angola system by the Angola Current (AC), reducing nitrate concentration in the euphotic layer. At the same time, the AC brings CHLa into the euphotic layer by meridional advection, as the EUC has relatively high CHLa concentrations (Radenac et al., 2020). This low nitrate / high CHLa signature of the AC can be seen in the first hundred metres and in particular at

Our analysis reveals that the seasonal variability of CHLa in our region is driven by the seasonal

the base of the euphotic layer along the Congolese coast (Fig.A1c), where the AC flows (Kopte et al, 2017). Further analyses show that the coastal CHLa maximum occurs from May to September with a peak in August, which is consistent with the seasonal cycle of the CHLa concentration in the EUC (Radenac et al, 2020; Brandt et al, 2023). The simultaneous variation in current and gradient appears to be the main factor contributing to variations in the meridional advection and nitrate removal, mainly between July and October. Over the same period, we observe a sign change of the meridional nitrate gradient (Fig.15b), which is generally positive (indicating that under the mixed layer, waters to the south of the box are less nitrate-rich than waters to the north) under the mixed layer from January to June.

**Figure 17**: Regional distribution of mean annual nitrate and CHLa concentration (a,b respectively) averaged on the 0-100 m layer along the Congolese upwelling area, black line represents different boundaries of our Coastal box with a width of 1° of longitude relative to the coast.

This observation suggests that for the period from September to October, waters in the southern part of the box are now richer in nitrate than waters in the northern part of the box under the mixed layer. This reflects the passage of low-nitrate waters from the EUC via the Angola Current to the Congolese coast. However, we find that zonal advection brings nitrate in the euphotic layer and its decomposition has shown that its variation is mostly influenced by zonal nitrate gradient variation, with a secondary contribution of zonal current variation. This zonal current variation is seen to remove nitrate in the euphotic layer during February-March and July-October. We can also see that in these periods the zonal current flows toward the coast, which suggests that it brings low nitrate water from offshore toward the coast. This seasonally eastward current is consistent with the seasonal cycle of SEUC (Siegfried et al., 2019; Assene et al., 2022). Besides this, we can also see that there is a sign change of the zonal nitrate gradient (Fig.14b) which occurs simultaneously with the sign change of the meridional nitrate gradient (Fig.15b) suggesting that this inversion in zonal nitrate gradient is due to Angola current water at the coast. In fact, this later result highlights that the Angola current waters flowing along the coast are less rich in nitrate than the water brought from offshore to ward the coast by the SEUC. As nitrate concentration in the coast is lower, SEUC waters act to increase nitrate concentration in our coastal box,

through zonal advection in the euphotic layer from April to December. The CHLa budget analysis (Appendix A) shows that the SEUC brings through zonal advection low CHLa water at the coast thereby reducing the nitrate input during the downwelling period. In the nitrate budget we saw that the main driver of nitrate input was vertical advection associated with CTWs. Körner et al. (2024) have shown, using satellite and mooring data, that CTWs detected in the SLA are of the low vertical mode, while the movement of the isopycnals is rather consistent with the vertical velocity structure of higher modes. This explains why isopycnals reach their seasonal minimum/maximum depth (in phase with the nitracline) after the minimum in SLA (Körner et al, 2024).

We assessed the variability of biological productivity using the PISCES component of our coupled model (Fig.18), and the results are similar to the semi-annual cycle of nitrate and CHLa concentration in the TAUS (Korner et al., 2024). The highest values of total primary production (TPP) are observed near the mouth of the Congo River (between 5-6°S), with more than 0.20 molN.m<sup>-2</sup>.d<sup>-1</sup> on average. The maximum value of NPP occurs during the main upwelling period, with 0.60 molC.m<sup>-2</sup>.d<sup>-1</sup>, which is almost four times less than the average primary production in the Benguela and Humbolt upwelling systems (2.49 and 2.18 molC.m<sup>-2</sup>.d<sup>-1</sup> respectively) and three times less than that in the Canary Islands upwelling system (Monteiro et al., 2010). In the secondary upwelling period, NPP reaches around 0.26 molN.m<sup>-2</sup>.d<sup>-1</sup>, which is more than twice lower than in the main upwelling period. In the downwelling period, the net primary production is less than 0.20 molN.m<sup>-2</sup>.d<sup>-1</sup>, whereas in the main upwelling period, the new production is estimated at around 0.30 molN.m<sup>-2</sup>.d<sup>-1</sup>, which represents around 50% of the net primary production with 0.07 molN.m<sup>-2</sup>.d<sup>-1</sup>.

**Figure 18**: Latitude-Time Hovmöller diagram of biological productivity: Net Primary Production (a), New Production (b) and regenerated Production (c) in the coastal box (6°S-3°S, 1° wide along the coastline). The units are milli mol per cubic meter per day.

773774

779

781

791

794

797

800

804

806

We also have the highest values of TPP (Net primary production) at 4.47°S (mouth of the Kouilou), in line with the semi-annual cycle. We can also see that regenerated primary production (fig.18c) (the difference between Net TPP and NP), due to recycling of nitrate in the euphotic layer, appears to be greater than new production due to nitrate from outside the euphotic layer. The above two parameters are consistent with the seasonal cycle of nitrate concentration. Net primary production in our study area has a different seasonal cycle than in Namibia and Benguela, as these are wind-forced upwelling areas (Gutknecht, 2013), unlike our study area. Another important observation is that the contribution of new production to net primary production in our area is larger than in wind upwelling systems. For example, in the Benguela upwelling new production contributes about 30% of net primary production, with the f-Ratio between 0.2 and 0.4 (Monteiro et al., 2010), whereas it is about 0.6 in our study area. We also note that conclusions about the nitrate budget in the mixed layer can vary depending on the definition of the mixed layer, particularly when we look at the main process driving nitrate input to the mixed layer. As mentioned earlier in this study, the main driving process of nitrate input to the mixed layer is vertical advection when we take a mixed layer close to 10 m depth (de Boyer Montégut criteria). But in the Angolan upwelling system, several recent studies (Zang et al., 2023, Körner et al., 2023, 2024, Brandt et al., 2023) mention that due to vertical mixing induced by the internal tides, turbulent mixing is the main driver of coastal cooling in this zone and that this conclusion is probably the same in the Congolese coastal zone. Thus, in our analysis, if we consider a very shallow mixed layer using the same definition of mixed layer as Ngakala et al. (2025) and Scannel and Mc Phaden (2018) in the Congolese zone, we arrive at the same conclusion as previously found in the Angolan coastal zone (Zang et al., 2023, Korner et al., 2023, 2024, Brandt et al., 2023) with the dominance of vertical diffusion in the nitrate budget.

5 Conclusion

Throughout this work we have described and analysed the seasonal cycles in nitrate and CHLa concentrations, as well as the physical and biological processes that modulate nitrate supply and biological productivity in the mixed layer and in the euphotic layer in the Congolese upwelling system. We began by validating a regional high-resolution (1/36°) simulation of the coupled physical-biogeochemical model NEMO-PISCES in this area for the studied year 2011. Surface and subsurface validation of the simulation using observations (satellite, in situ, climatology) shows that the model reasonably reproduces the main physical and biogeochemical characteristics of the study area. Subsequently, the seasonal cycle of nitrate shows that there are two periods of upwelling (May-August and December) and two periods of downwelling (January-April and October-November). These upwelling and downwelling are associated with remote forcing: Kelvin waves that propagate along the equator and the coastal waveguide force the vertical migrations of the thermocline, which is also a proxy for the nitracline. The seasonal cycle of CHLa is explained by that of nitrate. The assessment of the nitrate balance in the mixed layer shows that the main nitrate is mainly supplied in the mixed layer by

812

814

815816

817818

zonal advection, which is mainly modulated by nitrate inputs from the Congo River at 6°S. The vertical advection induced by CTWs and vertical diffusion play also a role in the nitrate supply, while nitrate losses are linked to meridional advection and the biological activity (photosynthesis). In the lower part of the euphotic layer, on the other hand, nitrate is supplied by zonal advection and vertical advection. Vertical diffusion contributes to nitrate losses, except in downwelling periods where it represents one of the main drivers of nitrate supply. We have also seen that meridional advection via the Angola Current, which transports the low-nitrate warm waters of the Equatorial undercurrent, is the main driver of nitrate loss below the mixed layer throughout the year. We find that vertical advection is controlled by the vertical nitrate gradient and nitrate input, rather than vertical velocity, when it brings nitrate into the mixed layer during the main upwelling period. However, in the secondary upwelling in December, vertical advection also brings nitrate, but is then mostly controlled by vertical velocity. In future works, the interannual variability will be study especially associated with the interannual variability of the Congo river discharges (Scannell and McPhaden (2018), Körner et al., 2023, 2024; Brandt et al., 2023) and of the CTWs forced by the Equatorial Kelvin waves (e.g. Bachèlery et al., 2015, 2016). Understanding the seasonal and interannual variability of productivity is of primary interest to ensure the sustainability of ecosystems and fisheries in the Congolese upwelling system.

#### Appendix A: Euphotic Layer CHLa Budget Analysis

**Figure A1**: Depth-time Hovmöller diagram of the model seasonal cycle of horizontal and vertical process contributions (a and b respectively), Zonal, Meridional, Vertical advections (c, e and d respectively), Vertical diffusion (f) of CHLa in the Congolese coast. Units are milligram per cubic meter per day for all of the plots.

The strong black line represent seasonal variation of thermocline, dashed magenta line is the euphotic layer and dashed black line is mixed layer dept.

# 831 Appendix B: Euphotic Layer Nitrate Advections Components Analysis

Figure B1: Seasonal variation of advection components averaged in the euphotic layer, black line represent zonal, meridional and vertical advection in (a), (b) and (c) respectively. The red line in both three figures represents gradient variation, blue line is current variation and magenta line represents the simultaneous variation of gradient and current.

836837

832833

872

Code and data availability. Publicly available datasets were used for this study. Chlorophyll data 839 (1998-2020) are from the Copernicus-GlobColour dataset (https://doi.org/10.48670/moi-00281, 840 Copernicus, 2023a). The sea level anomaly data (1998–2020) were accessed via the Copernicus Server (https://doi.org/10.48670/moi-00148, Copernicus, 2023b). The MUR SST product created by the JPL 841 MUR MEaSURES program as part of the GHRSST (Group for High-Resolution Sea Surface 842 Temperature) project is obtained from <a href="https://podaac.jpl.nasa.gov/dataset/MUR-JPL-L4-GLOB-v4.1">https://podaac.jpl.nasa.gov/dataset/MUR-JPL-L4-GLOB-v4.1</a> 843 (Chin et al., 2017) and ASCAT wind data https://podaac.jpl.nasa.gov/dataset/ASCATB-L2-25km. The 844 nutrient fields were assessed using the CSIRO Atlas of Regional Seas climatology (Dunn and Ridgway, 845 846 847 https://thredds.aodn.org.au/thredds/catalog/CSIRO/Climatology/CARS/2009/AODN-848 product/catalog.html?dataset=CSIRO/Climatology/CARS/2009/AODN-product/. Near surface currents from the Ocean Surface Current Analysis Real-time (OSCAR, Johnson et al. 2007), 849 850 https://podaac.jpl.nasa.gov/dataset/OSCAR L4 OC INTERIM V2.0. Model outputs are available from the authors, especially GA, ID, GM, and JJ. 851 852 853 Author contributions. LJME outlined and wrote the paper. LJME and RDG produced the figures. GM 854 has run the NEMO-PISCES model. GA, ID, CYD the co-authors contributed to define methodology and 855 reviewed the paper. 856 **Conflict of Interest.** The authors declare that they have no conflict of interest. Acknowledgments. Research is sponsored by the CNES SWOT-ETAO project (Surface Water and 857 858 Ocean Topography, Study of Ocean Topography and Altimetry by the National Centre for Space Studies France). GENCI GEN7298 project (National High-Performance Computing Equipment) for computing 859 hours for simulations. 860 861 Thanks go to (https://www.esr.org/research/oscar/oscar-surface-currents/) to provide OSCAR current 862 Also thanks to <a href="https://podaac.jpl.nasa.gov/">https://podaac.jpl.nasa.gov/</a> for providing MUR SST data. 863 864 Financial Support. This project is funded by IRD-ARTS (Research Grant for a Thesis in the South 865 provided by Institute of Research for Development France) for my Phd scholarship. 866 References 867 Assene, F., Morel, Y., Delpech, A., Aguedjou, M., Jouanno, J., Cravatte, S., Marin, F., Ménesguen, C., 868 Chaigneau, A., Dadou, I., Alory, G., Holmes, R., Bourlès, B., & Koch-larrouy, A. (2020). From Mixing 869 to the Large Scale Circulation: How the Inverse Cascade Is Involved in the Formation of the Subsurface Currents in the Gulf of Guinea. 1-36. https://doi.org/10.3390/fluids5030147. 870

Aumont, O. and Bopp, L.: Globalizing results from ocean insitu iron fertilization experiments, Global

Biogeochem. Cy., 20, GB2017, https://doi.org/10.1029/2005GB002591, 2006.

- Aumont, O., Ethé, C., Tagliabue, A., Bopp, L., and Gehlen, M.: PISCES-v2: an ocean biogeochemical
- model for carbon and ecosystem studies, Geosci. Model Dev., 8, 2465–2513,
- https://doi.org/10.5194/gmd-8-2465-2015, 2015.
- Awo, F. M., Rouault, M., Ostrowski, M., Tomety, F. S., Da-Allada, C. Y., and Jouanno, J.: Seasonal
- cycle of sea surface salinity in the Angola Upwelling System, J. Geophys. Res.-Oceans, 127,
- e2022JC018518, https://doi.org/10.1029/2022JC018518, 2022.
- Bachèlery, M. L. (2016). Variabilité côtière physique et biogéochimique en Atlantique Sud-Est: rôle du
- forçage atmosphérique local versus téléconnexion océanique (Doctoral dissertation, Ph. D. thesis,
- Toulouse: Laboratoire d'Etude en Geophysique et Océanographie Spatiale (LEGOS), University of Paul
- Sabatier, 215).
- Brandt, P., Alory, G., Awo, F. M., Dengler, M., Djakouré, S., Imbol Koungue, R. A., Jouanno, J.,
- Körner, M., Roch, M., and Rouault, M. (2023). Physical processes and biological productivity in the
- upwelling regions of the tropical Atlantic. Ocean Science, 19(3):581–601, https://doi.org/10.5194/os-
- 19-581-202 3
- Caniaux, G., Giordani, H., Redelsperger, J. L., Guichard, F., Key, E., and Wade, M.: Coupling between
- the Atlantic cold tongue and the West African monsoon in boreal spring and summer, J. Geophys. Res.-
- Oceans, 116, C04003, https://doi.org/10.1029/2010jc006570, 2011.
- Carr, M.-E. (2002). Estimation of potential productivity in Eastern Boundary Currents using remote
- sensing. Deep Sea Research Part II: Topical Studies in Oceanography, 49(1–3):59–80.
- Chavez, F. P., & Messié, M. (2009). A comparison of eastern boundary upwelling ecosystems. Progress
- in Oceanography, 83(1-4), 80-96.
- Chin, T.M, J Vazquez-Cuervo et E Armstrong (2017). "A multi-scale high-resolution analysis of global
- sea surface temperature". In: Remote sensing of environment 200, p. 154-169.
- De Boyer Montégut, C., Mignot, J., Lazar, A., & Cravatte, S. (2007). Control of salinity on the mixed
- layer depth in the world ocean: 1. General description. Journal of Geophysical Research: Oceans,
- 112(C6).
- Dorothee Bonhoure, Elizabeth Rowe, Arthur J. Mariano, Edward H. Ryan. "The South Equatorial Sys
- Current." Ocean Surface Currents.(2004). https://oceancurrents.rsmas.miami.edu/atlantic/south-
- equatorial.html
- Dossa, A., Da-Allada, C., Herbert, G., & Bourlès, B. (2019). Seasonal cycle of the salinity barrier layer
- revealed in the northeastern Gulf of Guinea. African Journal of Marine Science, 41(2), 163-175.
- https://doi.org/10.2989/1814232X.2019.1616612
- Ducet, N., Le Traon, P.-Y., & Reverdun, G. (2000). Global high-resolution mapping of ocean circulation
- from TOPEX/Poseidon and ERS-1 and -2. Journal of Geophysical Research, 105(C819), 19477–19498.
- <u>https://doi.org/10.1029/2000jc900063</u>
- Estival, R., Quiniou, V., Messager, C., 2013. Real-time network of weather and ocean stations: public-
- private partnership on in-situ measurements in the Gulf of Guinea. Sea Technol. 54 (3), 34–38.
- FAO: Fishery and Aquaculture Country Profiles, Angola, 2020, Country Profile Fact Sheets, Fisheries
- and Aquaculture Division [online], Rome, https://www.fao.org/fishery/en/facp/ago?lang= en (last
- access: 11 April 2023), updated 7 February 2022.
- Fréon, P., Barange, M., & Arístegui, J. (2009). Eastern boundary upwelling ecosystems: integrative and
- omparative approaches. Progress in Oceanography, 83(1-4), 1-14.

- Gent, P. R. and McWilliams, J. C.: Isopycnal mixing in ocean circulation models, J. Phys. Oceanogr.,
- 20, 150–155, 1990.
- Gutknecht, E., Dadou, I., Marchesiello, P., Cambon, G., Le Vu, B., Sudre, J., Garçon, V., Machu, E.,
- Rixen, T., Kock, A., Flohr, A., Paulmier, A., and Lavik, G. (2013). Nitrogen transfers off Walvis Bay:
- a 3-D coupled physical/biogeochemical modeling approach in the Namibian upwelling system.
- Biogeosciences, 10(6):4117–4135.
- Hopkins, J., Lucas, M., Dufau, C., Sutton, M., Sturn, J., Lauret, O., & Channelliere, C. (2013). Detection
- and variability of the Congo River plume from satellite derived sea surface temperature, salinity, ocean
- colour and sea level. Remote Sensing of Environment, 139, 365-385.
- https://doi.org/10.1016/j.rse.2013.08.015
- Hutchings, L., van der Lingen, C. D., Shannon, L. J., Crawford, R. J. M., Verheye, H. M. S.,
- Bartholomae, C. H., van der Plas, A. K., Louw, D., Kreiner, A., Ostrowski, M., Fidel, Q., Barlow, R.
- G., Lamont, T., Coetzee, J., Shillington, F., Veitch, J., Currie, J. C., and Monteiro, P. M. S.: The
- Benguela Current: An ecosystem of four components, Prog. Oceanogr., 83, 15-32,
- https://doi.org/10.1016/j.pocean.2009.07.046, 2009.
- Johnson, E. S., Bonjean, F., Lagerloef, G. S., Gunn, J. T., & Mitchum, G. T. (2007). Validation and
- error analysis of OSCAR sea surface currents. Journal of Atmospheric and Oceanic Technology, 24(4),
- 688-701.

- Kopte, R. (2017). The Angola Current in a Tropical Seasonal Upwelling System: Seasonal Variability
- in Response to Remote Equatorial and Local Forcing (Doctoral dissertation, Christian-Albrechts
- Universität Kiel).
- Körner, M., Brandt, P., and Dengler, M.: Seasonal cycle of sea surface temperature in the tropical
- Angolan Upwelling System, Ocean Sci., 19, 121–139, https://doi.org/10.5194/os-19-121-2023, 2023.
- Körner, M., Brandt, P., Illig, S., Dengler, M., Subramaniam, A., Bachèlery, M. Lou, and Krahmann,
- G.: Coastal trapped waves and tidal mixing control primary production in the tropical Angolan
- upwelling system, Sci. Adv., 10, 29–31, https://doi.org/10.1126/sciadv.adj6686, 2024.
- Locarnini, M. M., Mishonov, A. V., Baranova, O. K., Boyer, T. P., Zweng, M. M., Garcia, H. E., ... &
- Smolyar, I. (2018). World ocean atlas 2018, volume 1: Temperature.
- Loukos, H. and Mémery, L.: Simulation of the nitrate seasonal cycle in the equatorial Atlantic ocean
- during 1983 and 1984, J. Geophys. Res., 104, 15549–15573, 1999.
- Madec, G. and the NEMO System Team, 2024. NEMO Ocean Engine Reference Manual, Zenodo.
- <u>https://doi.org/10.5281/zenodo.1464816</u>
- Messié, M., & Chavez, F. P. (2015). Seasonal regulation of primary production in eastern boundary
- upwelling systems. Progress in Oceanography, 134, 1-18.
- Monod, J.: Recherches sur la Croissance des Cultures Bactériennes, Hermann, Paris, 1942.
- Monteiro, P., Dewitte, B., Scranton, M., Paulmier, A., and Van der Plas, A. (2011). The role of open
- ocean boundary forcing on seasonal to decadal-scale variability and long-term change of natural shelf
- hypoxia. Environmental Research Letters, (6):1–14.
- Ngakala, R. D., Alory, G., Da-Allada, C. Y., Dadou, I., Cardot, C., Morvan, G., ... & Baloïtcha, E.
- (2025). Seasonal mixed layer temperature in the Congolese upwelling system. *Journal of Geophysical*
- Research: Oceans, 130(1), e2023JC020528.

- Nieto, K., & Mélin, F. (2017). Variability of chlorophyll-a concentration in the Gulf of Guinea and its
- relation to physical oceanographic variables. *Progress in oceanography*, 151, 97-115.
- Nubi, O., Bourles, B., & Edokpayi, C. (2016). On the Nutrient distribution and phytoplankton biomass
- in the Gulf of Guinea equatorial band as inferred from In-situ measurements. Journal of Oceanography
- *and Marine Science*, 7(1), 1-11.
- Ostrowski, M., da Silva, J. C. B., and Bazik-Sangolay, B.: The response of sound scatterers to El Niño-
- and La Niña-like oceanographic regimes in the southeastern Atlantic, ICES J. Mar. Sci., 66, 1063–1072,
- https://doi.org/10.1093/icesjms/fsp102, 2009.
- Radenac, M.-H., Jouanno, J., Tchamabi, C. C., Awo, M., Bourlès, B., Arnault, S., and Aumont, O.:
- Physical drivers of the nitrate seasonal variability in the Atlantic cold tongue, Biogeosciences, 17, 529–
- 545, https://doi.org/10.5194/bg-17-529-2020, 2020.
- Ridgway, K. R., J. R. Dunn, and J. L. Wilkin (2002), Ocean interpolation by four-dimensional least
- squares—Application to the waters around Australia, J. Atmos. Oceanic Technol., 19, 1357–1375.
- Rouault, M.: Bi-annual intrusion of tropical water in the northern Benguela upwelling, Geophys. Res.
- Lett., 39, L12606, https://doi.org/10.1029/2012gl052099, 2012.
- Schott, F. A., Fischer, J., and Stramma, L.: Transports and pathways of the upper-layer circulation in
- the western tropical Atlantic, J. Phys. Oceanogr., 28, 1904-1928, https://doi.org/10.1175/1520-
- 0485(1998)028<1904:TAPOTU>2.0.CO;2, 1998.
- Siegfried, L., Schmidt, M., Mohrholz, V., Pogrzeba, H., Nardini, P., Böttinger, M., and Scheuermann,
- G.: The tropical-subtropical coupling in the Southeast Atlantic from the perspective of the northern
- Benguela upwelling system, Plos One, 14, e0210083, https://doi.org/10.1371/journal.pone.0210083,
- 2019.
- Sikhakolli, R., Sharma, R., Basu, S., Gohil, B. S., Sarkar, A., & Prasad, K. V. S. R. (2013). Evaluation
- of OSCAR ocean surface current product in the tropical Indian Ocean using in situ data. Journal of earth
- system science, 122(1), 187-199
- Sowman, M. and Cardoso, P.: Small-scale fisheries and food security strategies in countries in the
- Benguela Current Large Marine Ecosystem (BCLME) region: Angola, Namibia and South Africa, Mar.
- Policy, 34, 1163–1170, https://doi.org/10.1016/j.marpol.2010.03.016, 2010.
- Tchipalanga, P., Dengler, M., Brandt, P., Kopte, R., Macueria, M., Coelho, P., Ostrowski, M., and
- Keenlyside, N. S.: Eastern Boundary Circulation and Hydrography Off Angola: Building Angolan
- Oceanographic Capacities, B. Am. Meteorol. Soc., 99, 1589–1605, https://doi.org/10.1175/Bams-D-
- 17-0197.1, 2018a.
- Tilstone, G., Smyth, T., Poulton, A., and Hutson, R. (2009). Measured and remotely sensed estimates
- of primary production in the Atlantic Ocean from 1998 to 2005. Deep Sea Research Part II: Topical
- Studies in Oceanography, 56(15):918–930.
- Topé, G. D. A., Alory, G., Djakouré, S., Da-Allada, C. Y., Jouanno, J., & Morvan, G. (2023). How does
- the Niger River warm coastal waters in the Northern Gulf of Guinea? Frontiers in Marine Science, 10,
- 1187202. https://doi.org/10.3389/fmars.2023.1187202.
- Tuchen, F. P., Brandt, P., Lübbecke, J. F., and Hummels, R.: Transports and pathways of the tropical
- AMOC return flow from Argo data and shipboard velocity measurements, J. Geophys. Res.-Oceans,
- 127, e2021JC018115, https://doi.org/10.1029/2021JC018115, 2022a.
- Xu, Z., M. Li, C. M. Patricola, and P. Chang, 2014: Oceanic origin of southeast tropical Atlantic biases.
- Climate Dyn., 43, 2915–2930, https://doi.org/10.1007/s00382-013-1901-y.

https://doi.org/10.5194/egusphere-2025-5112 Preprint. Discussion started: 11 November 2025 © Author(s) 2025. CC BY 4.0 License.

| 1000<br>1001<br>1002 | Zeng, Z., Brandt, P., Lamb, K. G., Greatbatch, R. J., Dengler, M., Claus, M., and Chen, X.: Three-dimensional numerical simulations of internal tides in the Angolan upwelling region, J. Geophys. ResOceans, 126, e2020JC016460, https://doi.org/10.1029/2020JC016460, 2021. |
|----------------------|-------------------------------------------------------------------------------------------------------------------------------------------------------------------------------------------------------------------------------------------------------------------------------|
| 1003<br>1004         | Zweng, M. M., Seidov, D., Boyer, T. P., Locarnini, M., Garcia, H. E., Mishonov, A. V., & Smolyar, I. (2019). World ocean atlas 2018, volume 2: Salinity.                                                                                                                      |