# Peer review of "Physical and biological processes driving seasonal variability of Nitrate"

_EGUsphere, 2025_

## Referee Comment (RC2)

**Review comments:**

Review of Landry Junior Mbang Essome et al. submitted to Ocean science.
"Physical and biological processes driving seasonal variability of Nitrate budget and biological productivity in the Congolese upwelling system."

The present manuscript is aimed at describing the relative contributions of physical and biological processes in modulating nitrate concentrations in the Congolese upwelling system. Although, I found the manuscript quite well written, there are some points that need to be clarified (see my comments). Thus, I recommend major revisions to publish this manuscript.

General comments:

The paper is generally well written. However, the authors should provide more details and clearer descriptions in several sections (see comments below). I also find it unusual that the authors consistently use the present tense in the Introduction when referring to past studies. I would suggest using either the present perfect or past simple instead.

In the Discussion and Conclusion sections, I recommend that the authors explicitly cite figures when recalling key results. This would improve the overall readability of the manuscript and be beneficial for readers. In addition, I suggest moving Figure 3 as a subplot within Figure 1 in order to better highlight the study area in the Introduction, and removing the red box currently shown in Figure 1.

The choice of the study area also requires clarification. Why did the authors restrict the domain to between 6°S and 3°S? Based on Figures 1 and 3, the northern boundary could be extended to 0°N, especially since the authors state that the region between 6°S and 0°N is poorly documented (L144). I would suggest that the authors extend the northern boundary to 0°N.

Regarding the mixed-layer nitrate budget, I can understand the assumption that lateral diffusion term is negligible; however, what about the entrainment term at the base of the mixed layer? This contribution has not been shown or discussed in the analysis and should be addressed.

To avoid confusion, I recommend removing the acronyms NPP, PP, TPP, NP, RP, as well as AC, and using the full terms throughout the manuscript. These acronyms are used only sparsely and, in several instances, incorrectly (e.g., total PP in L281, new primary production (NPP) in L285, regenerated production (RPP) in L286, or in the title of Figure 18 where NP is used for new primary production, RP for regenerated production, and TPP for total primary production).

It would also be helpful if the authors clearly define how the standard deviation and correlation coefficients were calculated in the Methods section.

Almost all figures need to be revised. Many figures lack clear boundaries or frames, and colorbars are often too close to the y-axis (e.g., Figures 9, 11, 13), making it difficult to distinguish between latitude/depth and colorbar values. Figures and colorbars should be clearly separated. In addition, I suggest using a finer color scale (more color levels) to better highlight differences.

From Figures 8, 10, and 12, it appears that there may be a discrepancy between the summed contributions of physical and biological processes and the total nitrate concentration tendency. Could the authors clarify this point? Is there a residual term in the budget?

Some figures are not discussed anywhere in the manuscript (e.g., Figures 11c, 11e, 17, and Appendix B), and this should be corrected. I also suggest that the authors consistently use either "Figure" or "Fig." throughout the manuscript when referring to figures.

Finally, there is a substantial number of missing references, both in the text and in the reference list, which must be addressed.

Other comments:

L3: I suggest that the list of affiliations start with the first author's affiliation.

L79: CTWs abbreviation should be used first in L76.

L111 and L777: It should be: "…(*Gutknecht et al., 2013…*" otherwise, this reference is missing in the reference section.

L115: I would suggest to add "South" after "eastern boundary of the …"

L123: It should be: "… the Angolan coast"

L134: What do you mean by twhich? Did the authors want to write which?

L180: It should be: "The red box indicates…".

L229: It should be: "…surface wind vector SST dataset. They …"

L247-250: The authors should rewrite these sentences using the correct definitions for each term in the budget.

L274: Equation 4 is missing in the manuscript.

L280: It should be: "… availability. Note …"

L292: SSH not defined. Also, be consistent between SLA and SSH throughout the manuscript.

L293: I would suggest: "… regional distributions from observations …"

In Figures 2a-b, what could explain the warm bias in the model west of 10ºE?

L330-331: What could explain the strong nutrient depletion in the model compared to the observations in surface waters, as shown in Figure 4? In addition, Figure 4 shows that the maximum vertical nitrate gradient is shallower and more pronounced in the model than in the observations. Could the authors elaborate on this difference? Does it have any implications for the results or their interpretation?

The different parameters and panels described in the caption of Figure 5 do not correspond correctly to the figure. I suggest that the authors carefully check and revise this caption.

The x-axis of Figure 6 seems to be cropped.

L445: It should be: "… (a) and …"

Spaces are needed between the figures and the different figures titles in Figures 9, 14, etc… .

L469-471: No possible to see with the actual colobars for Figures 10c-d.

L480: It should be: "… to the processes …"

Are the euphotic layer depth and the nitrate budget in the euphotic layer computed online or offline? I suggest that the authors comment on that in the method section.

L519: It should be: "…period (June, July, …)"

L526: I would suggest to remove "in the euphotic layer". It should be: "… (b) contributed by physical processes (c) and biological processes (d) …"

L543: I suggest: "The seasonal cycle of zonal nitrate advection ..." instead of "The zonal nitrate advection seasonal cycle …"

L544: "$<$" is missing. It should be "$< u >$". Also Is the correlation (r=0.77) statistically significant? If yes, at which confidence level? I suggest to mention it for the other correlation coefficients too.

L547: It should be: "… the term. The …"

L551: I suggest: "… Undercurrent (SEUC), …"

I suggest to write "zonal current", "meridional current" and "verticial current" instead of "current" in the captions of Figures 14, 15, 16, respectively.

L567 and L570: I suggest to write "meridional current" instead of "current"

L585: Rephrase the sentence since it is not only the vertical velocity variation term which is expressed in brackets.

L607: By using "consider" at the beginning of the sentence, do you mean "average". If yes, I sggest: "However, if we average in the upper 100 m, …"

L620: It should be: "… budget drivers in the Congolese upwelling system …"

L628: It should be: "… surface currents, …"

L697: The year is missing in the reference *Brandt et al*.

L744: It should be: "toward" instead of "to ward".

L758-766: I am confused by the values and units used to describe primary production, which sometimes do not appear to be consistent with Figure 18, unless a colorbar is missing. For example, a maximum NPP value of 0.60 mol C m$^{-2}$ d$^{-1}$ is mentioned. What does NPP refer to here? Is this variable shown in Figure 18? If so, a corresponding colorbar with the appropriate unit (mol C m$^{-2}$ d$^{-1}$) should be included, but it is currently missing. Moreover, the value of 0.60 does not appear to be visible in Figure 18. I suggest that the authors carefully revise this section, remove all misleading acronyms (including those in Figure 18), and thoroughly check the consistency of the units.

Figure A1: The figure is unclear, the y-axes are missing, and the description provided in the caption does not match the different panels shown.

References: Several references listed in the reference section are not cited in the text and should either be cited appropriately or removed. For instance:
- L871-872
- L890-891
- L896-898
- L902-904
- L915-916
- L948-949
- L950
- L968-969
- L989-919

Citations in the text that are not written in the reference section:

- L124: Bachèlery et al. (2015)
- L252: Körner (2023) L252
- L155: Scannell and McPhaden (2018)
- L189: Aumont et al. (1998)
- L198: Kobayashi et al. (2015)
- L206: Thiam et al. (2024)
- L221: Dunn and Ridgway (2002)
- L245: de Boyer-Montégut et al. (2004)
- L365: Awo et al. (2023)
- L400: Bourlès et al. (2004)
- L655: Hardman-Mountford and McGlade (2002)
- L695: Jouanno (2010)
- L714: Johns et al. (2014)
- L785: Zang et al. (2023)